# Knowledge, attitudes and practices of community treatment supporters administering multidrug-resistant tuberculosis injections: A cross-sectional study in rural Eswatini

Ernest Peresu[1]*, J. Christo Heunis[2], N. Gladys Kigozi[2], Diana De Graeve[3]

1 Faculty of Economic and Management Sciences, Centre for Development Support, University of the Free State, Bloemfontein, South Africa, 2 Centre for Health Systems Research & Development, University of the Free State, Bloemfontein, South Africa, 3 Faculty of Applied Economics, University of Antwerp, Prinsstraat, Antwerp, Belgium

* eperesu@yahoo.com

**Data Availability Statement:** All relevant data are within the manuscript and its Supporting Information files.

## Abstract

### Background

This study assessed knowledge, attitudes and practices (KAP) of lay community treatment supporters (CTSs) delegated with directly observed treatment (DOT) supervision and administration of intramuscular multidrug-resistant tuberculosis (MDR-TB) injections in the Shiselweni region in Eswatini.

### Methodology

A cross-sectional survey among a purposive sample of 82 CTSs providing DOT and administering injections to MDR-TB patients was conducted in May 2017. Observations in the patients' homes were undertaken to verify CTSs' self-reported community-based MDR-TB management practices.

### Results

Out of 82 respondents, 78 (95.1%) were female and half (n = 41; 50.0%) had primary education or lower. Over one-tenth (n = 12; 14.6%) had not attended a MDR-TB training workshop, but were administering injections. The overall KAP scores were satisfactory. Good self-reported community-based MDR-TB practices were largely verified through observation. However, substantial proportions of respondents incorrectly defined MDR-TB, were unaware of the treatment regimen, stigmatised patients, and underreported needlestick injuries. There was no statistically significant association between duration administering intramuscular injections, MDR-TB training, knowledge and attitudes, and good community-based MDR-TB management practices.

**Funding:** The author(s) received no specific funding for this work.

**Competing interests:** The authors have declared that no competing interests exist.

## Conclusions

The gaps in the current KAP of CTSs in this setting raise questions about the timing, adequacy, design and content of community-based MDR-TB management training. Nonetheless, with appropriate training, lay CTSs in this region can be an option to complement an overstretched professional health workforce in providing DOT and MDR-TB injections at community level.

## Introduction

Multidrug-resistant tuberculosis (MDR-TB) is a major public health concern that is threatening global TB control. MDR-TB is defined as strains of TB resistant to the two most effective first-line anti-TB drugs, isoniazid and rifampicin [1]. The disease is often a consequence of inappropriate or interrupted treatment of drug-susceptible TB. In 2018, in Eswatini an estimated 509 patients were notified to have confirmed drug-resistant (DR-TB) compared to 110 DR-TB patients in 2008; that is more than a quadruple increase over a decade [2,3]. The rural areas in Eswatini carry a disproportionately high burden of MDR-TB. This can likely be ascribed to the primary health clinics (PHC) often being far away and geographically inaccessible from patients' home. Rural health services are also typically characterised by a lack of front-line TB human resources for health (HRH) [4,5].

The control of TB has been highlighted as a priority in the post-2015 global TB strategy (the End TB Strategy) and the Sustainable Development Goals agendas [1,6]. In 2018, Eswatini reported a 66% treatment success rate for MDR-TB patients (2016 cohort), that is, well below the World Health Organisation (WHO) target of 75% or higher [3,7]. As a coping mechanism to address MDR-TB treatment access and HRH challenges in the predominantly rural Shiselweni region, Médecins Sans Frontières (MSF) established a community-based MDR-TB treatment model in 2008. The main feature of this model is the task-shifting of directly observed treatment (DOT) supervision and MDR-TB injection administration responsibilities traditionally restricted to professional nurses, to incentivised trained lay community members within the existing National TB Control Programme (NTCP) [4].

In this model of care, MDR-TB patients discharged from the MDR-TB inpatient hospital are linked to a local lay community member known as a community treatment supporter (CTS), of their choice. The selected neighbour must have sufficient literacy skills to be able to comprehend the English training manual and document administered injections. Instead of making trips to the, MDR-TB patients receive their daily injections and DOT from CTSs in their (the patients') homes. Core clinical decisions such as monitoring of MDR-TB treatment and progress remain the purview of formal facility-based professional healthcare workers.

CTSs are typically recruited and trained to focus solely on MDR-TB. Their training, coordinated by community MDR-TB nurses, comprise on-the-job practical learning followed by a 3–5 days theoretical workshop at a later stage. During the first component of the training, the CTSs first practice safe injection techniques on orange fruits before administering injections to their patients. Before starting to administer injections on their own, CTSs should complete at least three observed injections assessed by a community MDR-TB nurse. Considerable emphasis is placed on drawing the right dose of the injectable drug, TB infection prevention and control (IPC), DOT and waste disposal. Newly recruited CTSs wait for up to a month to attend the theoretical component of the training in groups of ten.

The second component of the training focuses primarily on theoretical themes relating to MDR-TB epidemiology, transmission, diagnosis, treatment/strategies (DOT), safe injection handling, adverse drug reactions, and CTSs' responsibilities including patient confidentiality and procedures for patient referral to community MDR-TB nurses. The training is conducted by members of the formal health services through classroom instruction, group work and open discussions. Although the workshop uses pre- and post-training assessments, no certificates or any formal form of accreditation are given to CTSs on completion of the training. Each CTS receives a monthly stipend and irregular on-going supervisory visits from and at the discretion of the community MDR-TB nurse.

Although task-shifting in TB control is not a new concept in Eswatini, a key tension in the delegation of MDR-TB injection administration responsibilities to CTSs has been the absence of a national policy framework to regulate their recruitment, training and accreditation. Professional bodies have raised ethical concerns about standards of care and safety risks for both patients and CTSs ranging from potential errors in dosing to transmission of infections through unsafe injection handling and inappropriate community MDR-TB IPC practices [8]. As a result, there are fears that task-shifting injection administration to CTSs may create a two-tiered system of MDR-TB management, with "superior" and "inferior" tracks.

Previous extensive systematic literature reviews on task-shifting have established the safety, effectiveness and acceptability of using appropriately trained and supervised lay community health workers (CHWs) in the provision of injectable contraceptives [9] and delivery of vaccines and medicines to mothers and children through compact pre-filled injections [10]. The experience in this case study setting is important given the paucity of programmatic experience globally in the provision of MDR-TB injections by lay CHWs.

This study was conducted to assess the KAP levels of CTSs that may facilitate or impede the supervision of DOT and administration of injection treatment in the community-based MDR-TB programme. Results will help in formulating recommendations for optimising community-based MDR-TB management training for CTSs in the Shiselweni region.

## Methods

### Setting and design

The Shiselweni region, with an estimated population of 204 111 in 2017, has three main health facilities supporting 18 smaller PHC clinics that form part of the regional health network managed by the Ministry of Health [11]. CTSs accompany their MDR-TB patients to the main health facilities for their monthly outpatient treatment review and any unscheduled visits in the case of worsening health condition.

A cross-sectional survey was conducted among CTSs providing DOT and administering injections to MDR-TB patients in the Shiselweni region in May 2017. Direct observation of CTSs supervising DOT and administering intramuscular injections in the MDR-TB patients' homes was conducted and recorded on a structured checklist.

### Sampling

A purposive sample of 82 out of a study population of 124 CTSs enrolled in community-based MDR-TB management in the Shiselweni region was considered for the survey. The inclusion criteria were having at least one month experience of and currently administering MDR-TB injections. From a list of the 82 CTSs that participated in the survey, 20 were selected using a stratified sampling method to verify self-reported practices by direct observation. The sampling frame (list of CTSs) for each stratum was obtained from the register of CTSs at each MDR-TB treating facility. Within each stratum, participants were selected through

proportionate random sampling to reach a target sample size for each facility–Matsanjeni Health Centre (10), Nhlangano Health Centre (6) and Hlatikhulu Hospital (4). All eligible respondents that were approached agreed to participate in the survey and observations.

## Instrument development and measures

An interviewer-administered structured questionnaire consisting of 64 items (S1 File) was developed based on literature review and CTS training materials and job descriptions [12–16]. The content validity of the questionnaire was assessed by expert opinion. The internal reliability of the KAP scale was found to be acceptable and had a satisfactory discriminating power with subscale Cronbach's alpha co-efficient for knowledge, attitude and practice of 0.72, 0.68 and 0.62 respectively [17–21].

The questionnaire collected socio-demographic details as shown in Table 1. There were 25 knowledge-related questions that examined the definition of MDR-TB, its aetiology, transmission, main symptoms, diagnosis, treatment, safe injection handling, and IPC as shown in Table 2. Respondents were asked to choose the correct response on a given statement on a scale that ranged from 1 ("yes"), to 2 ("unsure"), and 3 ("no"). Responses to knowledge questions were assigned a score of 1 for correct and 0 for inappropriate or uncertain responses. CTSs' attitudes towards community-based MDR-TB management were measured using 21 statements. Participants chose either "strongly agree" (5 points), "agree" (4 points), "unsure" (3 points), "disagree" (2 points) and "strongly disagree" (1 point) on a 5-point Likert-type scale as shown in Table 3. CTS's self-reported practices regarding MDR-TB were assessed based on responses to 12 questions as shown in Table 4. The self-reported practices were verified by carrying out observations in a sample of 20 CTSs using a structured observation checklist, as shown in Table 5.

**Table 1. Socio-demographic characteristics.**

|  | N = 82 (%) |
|---|---|
| Sex |  |
| Male | 4 (4.9) |
| Female | 78 (95.1) |
| Age group |  |
| ≤ 30 years | 10 (12.2) |
| 31–40 years | 21 (25.6) |
| 41–49 years | 18 (22.0) |
| ≥ 50 years | 33 (40.2) |
| Education level |  |
| Primary school or lower | 41 (50.0) |
| Secondary school or higher | 41 (50.0) |
| Months administering MDR-TB injections |  |
| 1–4 months | 17 (20.7) |
| > 4 months | 65 (79.3) |
| Attended MDR-TB training workshop in the past 12 months |  |
| Yes | 70 (85.4) |
| No | 12 (14.6) |
| Received MDR-TB on-the-job training in the past 12 months |  |
| Yes | 82 (100.0) |
| No | 0 (0) |

**Table 2. CTSs' knowledge about MDR-TB.**

| CTSs knowledge items (correct response) | Correct response |
|---|---|
| | n (%) |
| MDR-TB are strains of TB resistant to at least isoniazid and rifampicin (yes) | 48 (58.5) |
| MDR-TB is contagious (yes) | 81 (98.8) |
| A CTS providing care to a patient with MDR-TB may develop MDR-TB (yes) | 79 (96.3) |
| People who sleep in the same room are not close TB contacts (no) | 36 (43.9) |
| Babies under two years are close TB contacts of their parents, or anyone who looks after them (yes) | 79 (96.3) |
| A person can get MDR-TB from shaking hands with someone with MDR-TB (no) | 61 (74.4) |
| A person with HIV is more likely to develop MDR-TB (yes) | 79 (96.3) |
| Opening windows can help in preventing the spread of MDR-TB (yes) | 82 (100) |
| Wearing a N95 respirator can reduce the risk of transmission of MDR-TB (yes) | 79 (96.3) |
| All people with MDR-TB infection have visible symptoms (no) | 68 (82.9) |
| Coughing is the most common symptom of MDR-TB (yes) | 53 (64.6) |
| MDR-TB is best diagnosed from a chest X-ray (no) | 39 (47.6) |
| The correct way of assessing MDR-TB treatment outcome is through sputum culture (yes) | 79 (96.3) |
| MDR-TB can be cured (yes) | 80 (97.6) |
| General antibiotics given at the health centre can cure MDR-TB (no) | 73 (89.0) |
| MDR-TB is best treated with the following drug combination: rifampicin, kanamycin and levofloxacin only (no) | 11 (13.4) |
| The standard length of injection treatment for a newly diagnosed case of MDR-TB is eight months (yes) | 76 (92.7) |
| Kanamycin is the drug that is used for injection during the intensive phase (yes) | 28 (34.2) |
| The duration of treatment for MDR-TB is between 18 and 24 months (yes) | 77 (93.9) |
| Sometimes people with MDR-TB do not get better because they do not take their medication (yes) | 80 (97.6) |
| Medications with visible contamination or breaches of integrity (e.g. cracks, leaks) should be discarded (yes) | 79 (96.3) |
| Swabbing before injections will minimise the pain during injection (no) | 36 (43.9) |
| Recapping of used needles can cause needlestick injuries (yes) | 76 (92.7) |
| Taking antiretroviral drugs as post-exposure prophylaxis (PEP) can reduce the rate of infection in healthcare workers exposed to HIV through needlestick injuries (yes) | 49 (59.7) |
| An infection or boil on the injection site is a side effect related to the injection that should be reported to the community MDR-TB nurse (yes) | 73 (89.0) |

The questionnaire was availed in both English and the local language siSwati. The instrument was pretested for practicality among 10 CTSs who were no longer administering MDR-TB injections and were excluded from the study. A structured observation checklist (S2 File) for verifying CTSs' self-reported community-based MDR-TB management practices was developed based on a schedule used by community MDR-TB nurses in supervising CTSs and literature review [22–25]. The checklist comprised of 26 items recording CTSs' TB IPC, DOT, and injection administration practices.

## Participant recruitment and data collection

CTSs accompanying their MDR-TB patients for their monthly review at the three MDR-TB treating facilities were informed about the research (interviews and observations) by the community MDR-TB nurse at the end of their consultation. The CTS was then referred to a trained research assistant stationed in a private room within the MDR-TB unit at the health centre. Three research assistants with previous experience in data collection were recruited for the study. A two-day training workshop provided an overview of the study, basic knowledge of

**Table 3. CTSs' attitude towards MDR-TB.**

| CTSs' attitude items | Concur | Unsure | Differ |
|---|---|---|---|
| | n (%) | n (%) | n (%) |
| **Awareness**<br>MDR-TB is a major public health threat in Eswatini | 61 (74.4) | 0 (0) | 21 (25.6) |
| I feel awareness of MDR-TB in my community is adequate | 58 (70.7) | 0 (0) | 24 (29.3) |
| Community awareness about MDR-TB is important in the control of the disease | 77 (93.9) | 2 (2.4) | 3 (3.7) |
| **Training**<br>I understand the importance of attending regular training on TB prevention | 82 (100) | 0 (0) | 0 (0) |
| I have enough information about community MDR-TB management | 56 (68.3) | 1 (1.2) | 25 (30.5) |
| **Patient education**<br>It is my responsibility to teach patients about TB prevention | 77 (93.9) | 2 (2.4) | 3 (3.7) |
| **Infection prevention and control**<br>Patients with known MDR-TB should be separated from HIV patients | 57 (69.5) | 3 (3.7) | 26.8 |
| Washing my hands before and after direct patient contact is a necessary part of my work | 79 (96.3) | 1 (1.2) | 2 (2.4) |
| I encourage adequate ventilation in the patient's home, regardless of weather conditions | 80 (97.6) | 1 (1.2) | 1 (1.2) |
| I use a N95 respirator even though it may be uncomfortable | 81 (98.8) | 0 (0) | 1 (1.2) |
| **Risk of acquiring MDR-TB**<br>I worry about acquiring active MDR-TB disease while at work | 71 (86.6) | 0 (0) | 11 (13.4) |
| I think I have a very low risk of acquiring MDR-TB from my patient | 63 (76.8) | 1 (1.2) | 18 (22.0) |
| I believe following safe injection practices can help reduce the risk of infectious adverse events in healthcare providers | 80 (97.6) | 2 (2.4) | 0 (0) |
| **Adherence**<br>I think it is difficult for patients with MDR-TB to understand they need to continue taking medication after they start feeling better | 77 (93.9) | 4 (4.9) | 1 (1.2) |
| I consider interruption of the MDR-TB treatment course to be a possible cause of worsening of symptoms | 80 (97.6) | 1 (1.2) | 1 (1.2) |
| I believe taking traditional medicine makes the treatment of MDR-TB difficult | 37 (45.1) | 1 (1.2) | 44 (53.7) |
| **Compassion and stigma**<br>I feel I should show compassion to my MDR-TB patient | 80 (97.6) | 2 (2.4) | 0 (0) |
| MDR-TB patients are to blame for their own condition | 29 (35.4) | 3 (3.7) | 50 (61.0) |
| I think MDR-TB patients are confronted with significant social stigma surrounding the disease | 49 (59.8) | 5 (6.1) | 28 (34.2) |
| My MDR-TB patient may not want other people to know that he/she has TB | 60 (73.2) | 3 (3.7) | 19 (23.2) |
| **Supervision**<br>My supervisor is easily accessible when I need help in managing my MDR-TB patient | 80 (97.6) | 0 (0) | 2 (2.4) |

MDR-TB and TB IPC, data collection instruments, and the procedure for obtaining voluntary informed consent from respondents. The entire data collection process was pilot-tested with the research assistants during a field practice visit at one MDR-TB treating facility. For the observation visits, the first author accompanied community MDR-TB nurses during their routine supervisory visits to MDR-TB patients' residences and verified CTSs' self-reported MDR-TB practices using the structured observation checklist.

Participation in the study was voluntary and not linked to the CTSs' job security. No rewards were offered for participating in the research. No personal identifying information

**Table 4. Self-reported CTSs' community based MDR-TB practices.**

| | n | % |
|---|---|---|
| **MDR-TB training manual** | | |
| Do you have a CTS MDR-TB training manual? (yes) | 61 | 74.4 |
| How often do you refer to the CTS MDR-TB training manual? (always/frequently) | 58 | 70.7 |
| **Community MDR-TB education and awareness** | | |
| Are you personally involved in educating patients or communities about MDR-TB? (yes) | 62 | 75.6 |
| How often do you provide information on MDR-TB? (always/frequently) | 55 | 67.1 |
| **Environmental IPC** | | |
| How often is cross ventilation implemented in the room your MDR-TB patient sleeps? (always/frequently) | 80 | 97.6 |
| **Administrative IPC** | | |
| Are there enough supplies such as soap and clean water to wash your hands at patient homes? (yes) | 81 | 98.8 |
| How often do you wash your hands before direct contact with a MDR-TB patient? (always/frequently) | 80 | 97.6 |
| How often do you wash your hands after direct contact with a MDR-TB patient? (always/frequently) | 82 | 100 |
| **Personal Protective Equipment** | | |
| How often do you wear a N95 disposable respirator when attending to an MDR-TB patient? (always/frequently) | 79 | 96.3 |
| **Safe injection handling practices** | | |
| How often do you use a clean needle and syringe to draw up and administer medication? (always/frequently) | 82 | 100 |
| How often do you immediately place needles and syringes in a sharps disposal container after administering an injection? (always/frequently) | 82 | 100 |

was collected and findings were reported anonymously using aggregate analysis. Verbal and written informed consent was sought from all MDR-TB patients and CTSs prior to their participation in the interviews and observations respectively.

## Analysis

Data was captured and cleaned in Epi Info 7 before being exported to Stata Version 14 for analysis (S3 File). Descriptive statistics were used to summarise the data. Total KAP scores on each of the scales were obtained from the composite scores and converted into percentages. Knowledge scores were classified as poor ($\leq$ 39.9%), moderate (40.0% - 69.9%) and good ($\geq$ 70.0%). Composite attitude scores $\geq$ 80% were considered as positive. Respondents with practice scale scores < 75% and $\geq$ 75% were considered to display poor and good community-based MDR-TB management practices respectively. Three relatively similar studies were used to inform cut-off points for good levels of MDR-TB KAP [12,16,26]. Results from direct observation on CTSs' MDR-TB practices were expressed as percentages.

Binomial logistic regression analysis was used to establish factors that were significantly associated with good community-based MDR-TB management practices. The level of statistical significance was considered at p value < 0.05 and 95% confidence interval (CI).

## Ethical clearance and authorisation

Ethical approval was obtained from the Scientific and Ethics Committee of Eswatini and the University of the Free State's Health Sciences Research Ethics Committee (IRB00006240). Authorisation of the study was granted by the NTCP and MSF.

**Table 5. Community based MDR-TB practices observed at patients' homes.**

| Checklist Item | Yes | |
|---|---|---|
| | n | % |
| **MDR-TB education and awareness** | | |
| CTS MDR-TB training manual available | 15 | 75.0 |
| Patient disclosed MDR-TB status to his/her family | 20 | 100 |
| All household members have been screened for MDR-TB | 20 | 100 |
| **DOT** | | |
| Patient swallowed MDR-TB medicine in the presence of the CTS | 12 | 60.0 |
| From the patient card, the CTS provided all injections and oral drugs | 20 | 100 |
| **Infection control** | | |
| Patient sleeps alone in a separate room | 20 | 100 |
| Patient's room has windows | 20 | 100 |
| Windows in patient's room open | 20 | 100 |
| CTS wearing N95 respirator | 20 | 100 |
| Patient wearing surgical mask | 0 | 0 |
| Adequate supply of soap and clean water to wash hands | 20 | 100 |
| **Safe injection handling technique** | | |
| Hands washed before procedure | 20 | 100 |
| New single needle and single syringe used | 20 | 100 |
| Vial checked for content, dose, and expiration date | 20 | 100 |
| Syringe filled with contents of the vial | 20 | 100 |
| Air expelled from syringe | 20 | 100 |
| Careful disposal of the drawing up needle from syringe and replacement with a fresh one | 20 | 100 |
| Exact site for injection located | 20 | 100 |
| Injection site disinfected with alcohol preparation pad | 20 | 100 |
| Patient advised to relax the muscle | 18 | 90.0 |
| Needle inserted swiftly at an angle of 90 degrees | 20 | 100 |
| Aspirated briefly to ensure the needle is not sited in a blood vessel | 18 | 90.0 |
| All contents of the syringe injected slowly (less painful) | 20 | 100 |
| Injection site gently pressed with a clean cotton ball | 20 | 100 |
| Needle and syringe disposed intact in a puncture-resistant sharps container | 20 | 100 |
| Hands washed after procedure | 16 | 80.0 |
| Information recorded on patient's card and other data collection forms | 20 | 100 |

# Results

Table 1 presents a summary of the demographic characteristics of the study sample. A large majority of participants (n = 78; 95.1%) were female and 33 (40.2%) were older than 50 years. Half (n = 41; 50.0%) of the respondents had low literacy and numeracy skills (primary education or lower) compared to the other half with secondary or higher education.

The length of service of respondents administering MDR-TB injections ranged from 1 to 8 months, with a mean duration of 5.8 months. More than one-tenth (n = 12; 14.6%) of newly recruited CTSs administering injections had not yet attended the 3–5 days theoretical training on community-based MDR-TB management.

## Assessment of CTSs' MDR-TB knowledge

The mean knowledge score for the respondents was 70.8% (standard deviation [SD]: ± 8.0%) with correct responses ranging from 40.2% to 88.2%. Overall, 71.9% and 28.1% of the CTSs

had good or moderate knowledge scores respectively. However, poor knowledge was apparent in responses to some individual questions relating to the definition, transmission, diagnosis and treatment of MDR-TB as well as safe injection handling, as shown in Table 2.

Less than two-thirds (n = 48; 58.5%) of the respondents correctly defined MDR-TB. Over four-fifths (n = 71; 86.6%) incorrectly identified the drug combination used in the treatment of MDR-TB. More than half (n = 46; 56.1%) incorrectly answered that swabbing before injections minimises pain during injection. However, a large majority of the CTSs (n = 73; 89.0%) correctly recognised injection site reactions as adverse effects of the intramuscular injection treatment that required timely referral.

## Attitudes

The mean attitude score of the CTSs was 93.5% (± 9.9%), ranging from 55.6% to 100% of appropriate responses. Overall, a large majority of respondents (n = 71; 86.6%) reported positive attitudes regarding community-based MDR-TB management. Despite these positive attitudes, just more than a quarter (n = 21; 25.6%) of CTSs did not consider MDR-TB to be a major public health threat in Eswatini, as shown in Table 3. More than half (n = 45; 54.9%) of the respondents believed that taking traditional medicines was not a hindrance to MDR-TB management. Almost nine in every ten (n = 71; 86.6%) respondents expressed fear of acquiring MDR-TB infection. More than one-third (n = 29; 35.4%) of the CTSs incorrectly believed that MDR-TB patients were themselves to blame for their condition.

## Practices

The mean practice score of CTSs was 83.9% (± 11.81%), ranging from 46.7% to 100% of good practice responses. Overall, a majority of respondents (n = 62; 75.6%) reported good practices regarding implementation of TB IPC measures, DOT and safe injection handling procedures as shown in Table 4. Some of these self-reported good practices were verified through observations (Table 5).

Self-reported environmental IPC practices were well applied and this was confirmed by the observation that all windows of the patients' rooms were open. A majority (n = 79; 96.3%) of respondents reported wearing a N95 disposable respirator when attending to a MDR-TB patient. However, none of the MDR-TB patients visited was observed wearing a surgical mask due to lack of supplies. Also one-fifth (n = 4; 20.0%) of the CTSs did not perform hand hygiene after patient contact.

The DOT card revealed that all CTSs consistently administered injections and oral medications daily. In the observations carried out, checking for sterility and expiry of syringes and vials; and the correct dosage, site and technique of injection was noted. Nine respondents (11.0%) reported having sustained a needlestick injury during practice. Of these, only two were reported and post-exposure prophylaxis (PEP) was reportedly not recommended. Reasons for not reporting needlestick injuries included that the injury occurred before administering the injection (n = 4); perceived low risk of infection from the injury (n = 1); lack of awareness about the need to take PEP (n = 1); and job security fears (n = 1).

## Predictors of good community-based MDR-TB management practice

Socio-demographic factors, i.e. duration administering injections, having attended MDR-TB training, sound MDR-TB knowledge and positive attitudes, were considered as potential explanatory variables for good community-based MDR-TB management practice. A bivariate logistic regression analysis (Table 6) found no significant association between individual socio-demographic variables such age, level of education, duration administering injections,

**Table 6. Bivariate and binomial logistic regression predicting community based MDR-TB management practice among CTSs.**

| Variables | Practice | | COR (95% CI) | p value | AOR (95% CI) | p value |
|---|---|---|---|---|---|---|
| | Good n (%) | Poor n (%) | | | | |
| **Age category** | | | | | | |
| ≤ 30 (Ref) | 6 (66.7) | 3 (33.3) | | | | |
| 31–40 | 17 (81.0) | 4 (19.0) | 2.13 (0.36–12.38) | 0.40 | | |
| 41–50 | 14 (73.7) | 5 (26.3) | 1.40 (0.25–7.83) | 0.70 | | |
| >50 | 25 (75.8) | 8 (24.2) | 1.56 (0.32–7.73) | 0.58 | | |
| **Education level** | | | | | | |
| Primary school or lower (Ref) | 30 (73.2) | 11 (26.8) | | | | |
| Secondary school or higher | 32 (78.0) | 9 (22.0) | 1.30 (0.47–3.59) | 0.61 | | |
| **Duration administering MDR-TB injections** | | | | | | |
| 1–4 months (Ref) | 10 (58.8) | 7 (41.2) | | | | |
| > 4 months | 52 (80.0) | 13 (20.0) | 2.8 (0.89–8.77) | 0.077 | 2.04 (0.38–1.12) | 0.41 |
| **Attended MDR-TB training workshop** | | | | | | |
| Yes (Ref) | 55 (78.6) | 15 (21.4) | | | | |
| No | 7 (58.3) | 5 (41.7) | 0.38 (0.11–1.38) | 0.14 | 0.21 (0.03–1.49) | 0.12 |
| **Knowledge** | | | | | | |
| Moderate (Ref) | 19 (82.6) | 4 (17.4) | | | | |
| Good | 43 (72.9) | 16 (27.1) | 0.57 (0.17–1.92) | 0.36 | 0.60 (0.17–2.16) | 0.43 |
| **Attitude** | | | | | | |
| Positive (Ref) | 32 (80.0) | 8 (20.0) | | | | |
| Negative | 30 (71.4) | 12 (28.6) | 0.63 (0.22–1.74) | 0.37 | 0.60 (0.21–1.76) | 0.36 |

Practice scores: Poor (< 75%), good (≥ 75%); COR: Crude odds ratio; AOR: Adjusted odds ratio; Ref: Reference.

MDR-TB training, knowledge and attitudes and good community-based MDR-TB management practice. A binomial logistic regression analysis revealed that after controlling for other variables in the model, none of the predictor variables were statistically significant.

## Discussion

This study contributes to extant knowledge about task-shifting by assessing the KAP of CTSs in community-based MDR-TB management in the Shiselweni region. While the overall level of CTSs' KAP relating to community-based MDR-TB management was satisfactory, the study identified some important gaps that deserve attention, in particular, the timing, adequacy, design and content of the training provided to CTSs.

The overall satisfactory KAP scores in this study masked substantial weaknesses in some aspects of CTSs' community-based MDR-TB management. Regardless of socio-demographic characteristics, some respondents were not aware of the definition of MDR-TB, its transmission, the rationale behind swabbing before administering an intramuscular injection and demonstrated unfamiliarity with drugs and regimens used in the treatment of MDR-TB. Similar results were found among healthcare workers in other countries [15,16,27,28]. Failure to identify medication names and fundamentals behind MDR-TB treatment care plan may adversely affect the ability of CTSs to recognise the doses, schedules and possible drug side effects during their interaction with patients.

Given CTSs' limited formal education and lack of familiarity with special medical terminology and technical concepts, the present study provides impetus for developing appropriately tailored training material on community-based MDR-TB management. The use of lay

conceptualisations and illustrations, and where possible, translated into the local language rather than using training packages developed for formal healthcare workers is recommended. The training course should integrate the theoretical and practical components to enhance a deeper understanding of community-based MDR-TB treatment concepts before delegating injection administration tasks to CTSs. Refresher training should be considered to be as important as the initial training.

The study findings highlighted salient erroneous and potentially stigmatising attitudes among CTSs. For instance, a substantial proportion of respondents inappropriately perceived that MDR-TB patients are to blame for their own condition. Similar to findings reported in previous studies among professional healthcare workers, most CTSs in the current study (86.6%) perceived themselves to have a high occupational risk of acquiring MDR-TB infection [26,29]. Studies from Ghana [30,31] and South Africa [32] have suggested that fear of infection is a major cause of TB stigma and can adversely influence patients' behaviours in accessing MDR-TB services [33–35]. These stigmatising attitudes should be addressed in the CTS training design to ensure the development of appropriate relationships between CTSs and their patients [36]. These findings also call for further qualitative research to assess the grounding for MDR-TB-associated stigma among CTSs.

The use of cross ventilation by opening windows and doors in this study was higher compared to observations in studies in Ethiopian (89.2%) and South African (69.0%) hospitals [37,38]. Nevertheless, of concern was the finding of low compliance with basic hand hygiene practice (after injection administration) among CTSs–similar to other studies [26,39]. Key strategies in this setting include the reinforcement of educational initiatives with written reminders for CTSs to recognise hand hygiene opportunities as well as the availability and use of low cost alcohol based hand rub to interrupt the cross-contamination chain [26,39–41].

The observations conducted, albeit limited in number, found the overall intramuscular injection practices of CTSs to be generally satisfactory. These findings were comparable to results reported among professional healthcare workers in a large Indian hospital [42] and CHWs administering injectable contraceptives [9,43]. In the current study, CTSs invariably administered injections and oral medications daily thereby optimising MDR-TB patients' adherence to treatment plans. Nevertheless, and similar to results from studies in Kenya [44] and Nigeria [45], of concern was the underreporting of needlestick injuries by CTSs. Needlestick injuries among healthcare workers can be considered as a cardinal indication of unsafe injection handling practices. Future research should identify strategies to reduce needlestick injuries and improve reporting among reporting-averse CTSs.

Without an enabling national policy on task-shifting, the results highlight potential patient safety and liability risks for the lay CHWs who undertake the delegated tasks [8]. With many health facilities stretched thin by the COVID-19 pandemic, task-shifting to appropriately trained, equipped and adequately supervised lay community members will optimise delivery essential TB treatment to communities. Apart from that, lay community members may play a critical role in facilitating, administering and expanding the reach of the COVID-19 vaccine. As such, there is need to develop conducive task-shifting guidelines to regulate the careful selection, appropriate training, accreditation and continuous supervision of CTSs.

In the present study, a majority of CTSs were female, indicating patients' gender preference in selecting their DOT and MDR-TB injection treatment provider. Research elsewhere has demonstrated that TB patients ought to pick the supporter of their inclination as selection of treatment supporter outside the health system does not adversely affect TB treatment outcomes [46]. Nevertheless, future studies could explore strategies that can strengthen gender equity in the recruitment of these frontline HRH and support male involvement in community-based MDR-TB care.

Converse to findings from previous research, there was no statistically significant correlation between individual socio-demographic variables, knowledge and attitudes and good community-based MDR-TB management practice [16,47]. Future research with a sufficiently larger sample is required to better establish the predictors of good community-based MDR-TB management practice.

With this study mostly founded on self-reported practice and compliance, social desirability bias may have occurred. However, this was partially countered by carrying out structured observations of CTS administering MDR-TB injections to complement survey responses. Respondents were also assured of anonymity of the questionnaires, reporting of findings using aggregate analysis and that the outcomes of the study would not affect their job security or incentives in any way. Although verification of self-reported practice was limited to 20 CTSs due to logistical feasibility, the observations provided the researchers with reasonable insights into CTSs' self-reported KAP and what they actually practice in community-based MDR-TB management.

## Conclusion

The study results indicate that in this setting there is the need for considerable strengthening of initial and on-going theoretical and in-service training programmes to reinforce awareness, address gaps in current knowledge and dispel misperceptions and potentially stigmatising attitudes regarding community-based MDR-TB management among CTSs. Taken together, these findings raise questions about the timing, adequacy, design and content of community-based MDR-TB management training provided to CTSs.

Although some problems remain, including the absence of a formal regulation framework for task-shifting and the limited literacy and numeracy skills of most CTSs in this study setting, trained lay CHWs can be an option to complement an overstretched health workforce in providing DOT supervision and administration of intramuscular MDR-TB injections at community level.

## Supporting information

**S1 File. Questionnaire.**
(DOCX)

**S2 File. Observation checklist.**
(DOCX)

**S3 File. Community treatment supporter survey.**
(XLS)

## Acknowledgments

The authors thank the NTCP and MSF for authorising the research. We also thank the community MDR-TB nurses and CTSs in the Shiselweni region for their support and participation in this study.

Availability of data and materials

The data analysed during this study are not publicly available as individual privacy would otherwise be compromised. Data is available on request from the corresponding author Ernest Peresu.

## Author Contributions

**Conceptualization:** Ernest Peresu, J. Christo Heunis, N. Gladys Kigozi, Diana De Graeve.

**Data curation:** Ernest Peresu.

**Formal analysis:** Ernest Peresu, J. Christo Heunis, N. Gladys Kigozi, Diana De Graeve.

**Investigation:** Ernest Peresu.

**Methodology:** Ernest Peresu, J. Christo Heunis, N. Gladys Kigozi, Diana De Graeve.

**Project administration:** Ernest Peresu.

**Supervision:** J. Christo Heunis, N. Gladys Kigozi, Diana De Graeve.

**Visualization:** Ernest Peresu.

**Writing – original draft:** Ernest Peresu.

**Writing – review & editing:** Ernest Peresu.

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
