## [Decision Letter · Decision Letter 0]

26 May 2022

PONE-D-22-07174Knowledge, attitudes and practices of community treatment supporters administering multidrug-resistant tuberculosis injections: a cross-sectional study in rural EswatiniPLOS ONE

Dear Dr. Peresu,

Thank you for submitting your manuscript to PLOS ONE. After careful consideration, we feel that it has merit but does not fully meet PLOS ONE’s publication criteria as it currently stands. Therefore, we invite you to submit a revised version of the manuscript that addresses the points raised during the review process. Please submit your revised manuscript by Jul 10 2022 11:59PM. If you will need more time than this to complete your revisions, please reply to this message or contact the journal office at plosone@plos.org. Please include the following items when submitting your revised manuscript:A rebuttal letter that responds to each point raised by the academic editor and reviewer(s). You should upload this letter as a separate file labeled 'Response to Reviewers'.A marked-up copy of your manuscript that highlights changes made to the original version. You should upload this as a separate file labeled 'Revised Manuscript with Track Changes'.An unmarked version of your revised paper without tracked changes. You should upload this as a separate file labeled 'Manuscript'.If applicable, we recommend that you deposit your laboratory protocols in protocols.io to enhance the reproducibility of your results. Protocols.io assigns your protocol its own identifier (DOI) so that it can be cited independently in the future. For instructions see: https://journals.plos.org/plosone/s/submission-guidelines#loc-laboratory-protocols. Additionally, PLOS ONE offers an option for publishing peer-reviewed Lab Protocol articles, which describe protocols hosted on protocols.io. Read more information on sharing protocols at https://plos.org/protocols?utm_medium=editorial-email&utm_source=authorletters&utm_campaign=protocols.

We look forward to receiving your revised manuscript.

Kind regards,

Bassey E. Ebenso, Ph.D., M.P.H., M.D.,

Academic Editor

PLOS ONE

Journal Requirements:

Additional Editor Comments (if provided):

Please address issues raised by reviewer 2 in the manuscript

Reviewers' comments:

Reviewer's Responses to Questions

**Comments to the Author**

1. Is the manuscript technically sound, and do the data support the conclusions?

Reviewer #1: Yes

Reviewer #2: Yes

2. Has the statistical analysis been performed appropriately and rigorously? 

Reviewer #1: Yes

Reviewer #2: Yes

3. Have the authors made all data underlying the findings in their manuscript fully available?

Reviewer #1: Yes

Reviewer #2: No

4. Is the manuscript presented in an intelligible fashion and written in standard English?

Reviewer #1: Yes

Reviewer #2: Yes

5. Review Comments to the Author

Reviewer #1: This is a very interesting and relevant study. The findings and recommendations on appropriate task-shifting of basic health services to lay community members, in resource-limited settings will be helpful for further research.

Author should complete the omission in the sentence, "Instead of making trips to the . . .., MDR-TB patients receive their daily injections and DOT from CTSs in their (the patients’) homes."

Reviewer #2: The authors’ address a topical subject of relevance to the target audience of PLOS ONE using quantitative methods to assess the knowledge, attitudes and practices of community treatment supporters administering multidrug-resistant tuberculosis injections in rural Eswatini. This should be encouraged. The authors should make minor revisions based on the following issues raised below:

1. Abstract

In the methodology section, authors did not include the number patient homes where observations were conducted on the CTSs actual practice.

Action: Authors to include the number of CTSs that were sampled for observation in the methodology section of the abstract.

2. Introduction

This paper is a deep dive into practical aspects of MDR-TB management and goes to length to describe the KAP of the CTSs on administering MDR-TB injections. The field of DR-TB management is a rapidly changing field as treatment regimen and recommendations are changed as new evidence is available. There is no description by the authors of the treatment regimen including type of injectable that was used in the country by the CTSs. This is critical information to help readers underground the program context/background better.

Action: Authors to include description of MDR-TB drug regimen in use at the time of the study as part of the background.

3. Methods:

a) In the sampling sub-section of the methodology, the authors describe how the 20 CTSs were selected through a proportionate random sampling from the 3 main facilities after a stratified sampling. The distribution of the 82 purposively selected CTSs among the 3 facilities which form the sampling frame for the stratified sampling is unclear.

Action: Clearly describe the distribution of the 82 purposively selected CTSs among the 3 main facilities. This will clarify the size of the sampling frame

b) In the Instrument development and measures sub-section, paragraph 2 (line 12) the authors describe Table 4 as having 12 questions assessing CTSs’ self-reported practices for MDR-TB. A review of table 4 shows responses to 11 questions rather than 12.

Action: Authors to review table 4 and update accordingly.

Thank you

6. PLOS authors have the option to publish the peer review history of their article (what does this mean?). If published, this will include your full peer review and any attached files.

Reviewer #1: No

Reviewer #2: No

---

## [Author Response · Author response to Decision Letter 0]

15 Jun 2022

REF: Revision and resubmission of manuscript PONE-D-22-07174

Dear Editor

Thank you for your email and the opportunity to revise our paper on ‘Knowledge, attitudes and practices of community treatment supporters administering multidrug-resistant tuberculosis injections: a cross-sectional study in rural Eswatini.’ The suggestions offered by the reviewer have been immensely helpful in improving some important aspects of the manuscript. 

We have included the academic editor and reviewers’ comments (in italics) immediately after this letter and responded to them individually, indicating exactly how we addressed each comment and describing the changes we have made. The revisions have been approved by all authors. The changes in the revised manuscript accompanying this letter are marked up.

We hope the revised manuscript will better suit PLOS ONE, but will be happy to consider further revisions and we thank you for your continued interest in our research.

Sincerely,

Ernest Peresu 

Academic Editor and Reviewer Comments and Author Responses 

Academic Editor

Comment 1: Please ensure that your manuscript meets PLOS ONE's style requirements, including those for file naming.

Response: Thank you for this reminder. We have revised our manuscript to ensure that it aligns with PLOS ONE's style requirements. 

Comment 2: Your ethics statement should only appear in the Methods section of your manuscript. If your ethics statement is written in any section besides the Methods, please delete it from any other section.

Response: We highly appreciate your comment. The ethics statement now only appears in the Methods section of our manuscript. 

Comment 3: We note that you have indicated that data from this study are available upon request. In your revised cover letter, please address the following prompts:

Response: Thank you for reminding us about the importance of making data from this study available. We have uploaded the minimal anonymised data set as a Supporting Information file (S3).

Comment 4: Please review your reference list to ensure that it is complete and correct.

Response: Thank you for this reminder and we have performed another thorough review of our reference list to ensure its both correct and complete. 

Reviewer 1

General comment: This is a very interesting and relevant study. The findings and recommendations on appropriate task-shifting of basic health services to lay community members, in resource-limited settings will be helpful for further research.

Response: We would like to thank Reviewer 1 for the thoughtful and thorough assessment of our manuscript. We are indeed grateful for the insightful comments.

Comment 1: Author should complete the omission in the sentence, "Instead of making trips to the . . .., MDR-TB patients receive their daily injections and DOT from CTSs in their (the patients’) homes."

Response: Thank you for pointing this out. The revised sentence now reads: ‘Instead of making trips to the clinics, MDR-TB patients receive their daily injections and DOT from CTSs in their (the patients’) homes’ (Page 4)

Reviewer 2

General comment: The authors’ address a topical subject of relevance to the target audience of PLOS ONE using both quantitative methods to assess the knowledge, attitudes and practices of community treatment supporters administering multidrug-resistant tuberculosis injections in rural Eswatini. This should be encouraged.

Response: We would like to thank the reviewer for carefully and thoroughly reading our manuscript and for the thoughtful comments and constructive suggestions. We are indeed grateful to the reviewer for his positive and encouraging comments. 

Comment 1: In the methodology section, authors did not include the number patient homes where observations were conducted on the CTSs actual practice. Authors to include the number of CTSs that were sampled for observation in the methodology section of the abstract.

Response: The methods section of the abstract has been improved by highlighting the number of CTSs that were sampled for direct observation (Page 2). 

Comment 2: Authors to include description of MDR-TB drug regimen in use at the time of the study as part of the background.

Response: We appreciate this comment and have accordingly added the description of MDR-TB drug regimen in use at the time of the study as part of the background (Page 4). 

Comment 3: In the sampling sub-section of the methodology, the authors describe how the 20 CTSs were selected through a proportionate random sampling from the 3 main facilities after a stratified sampling. The distribution of the 82 purposively selected CTSs among the 3 facilities which form the sampling frame for the stratified sampling is unclear. Clearly describe the distribution of the 82 purposively selected CTSs among the 3 main facilities. This will clarify the size of the sampling frame.

Response: We found this comment very helpful. We have rephrased and added more clarity on the distribution of the 82 CTSs among the 3 main facilities (Page 7). 

Comment 4: In the Instrument development and measures sub-section, paragraph 2 (line 12) the authors describe Table 4 as having 12 questions assessing CTSs’ self-reported practices for MDR-TB. A review of table 4 shows responses to 11 questions rather than 12. Authors to review table 4 and update accordingly.

Response: Thank you for pointing this out and we have reviewed the table accordingly. The sentence now reads: ‘CTS’s self-reported practices regarding MDR-TB were assessed based on responses to 11 questions as shown in Table 4’ (Page 9).

---

## [Decision Letter · Decision Letter 1]

29 Jun 2022

Knowledge, attitudes and practices of community treatment supporters administering multidrug-resistant tuberculosis injections: a cross-sectional study in rural Eswatini

PONE-D-22-07174R1

Dear Dr. Peresu,

We’re pleased to inform you that your manuscript has been judged scientifically suitable for publication and will be formally accepted for publication once it meets all outstanding technical requirements.

Kind regards,

Bassey E. Ebenso, Ph.D., M.P.H., M.D.,

Academic Editor

PLOS ONE

Additional Editor Comments (optional):

The revisions made to your manuscript (Revision Number 1) have sufficiently addressed comments raised by two independent reviewers

Reviewers' comments:

Reviewer's Responses to Questions

**Comments to the Author**

1. If the authors have adequately addressed your comments raised in a previous round of review and you feel that this manuscript is now acceptable for publication, you may indicate that here to bypass the “Comments to the Author” section, enter your conflict of interest statement in the “Confidential to Editor” section, and submit your "Accept" recommendation.

Reviewer #1: All comments have been addressed

Reviewer #2: All comments have been addressed

2. Is the manuscript technically sound, and do the data support the conclusions?

Reviewer #1: Yes

Reviewer #2: Yes

3. Has the statistical analysis been performed appropriately and rigorously? 

Reviewer #1: Yes

Reviewer #2: Yes

4. Have the authors made all data underlying the findings in their manuscript fully available?

Reviewer #1: Yes

Reviewer #2: Yes

5. Is the manuscript presented in an intelligible fashion and written in standard English?

Reviewer #1: Yes

Reviewer #2: Yes

6. Review Comments to the Author

Reviewer #1: We thank the author(s) for submitting the corrections and making necessary revisions. The corrections and comments have been satisfactorily addressed.

Reviewer #2: (No Response)

7. PLOS authors have the option to publish the peer review history of their article (what does this mean?). If published, this will include your full peer review and any attached files.

Reviewer #1: No

Reviewer #2: No

---

## [Editor Report · Acceptance letter]

5 Jul 2022

PONE-D-22-07174R1 

Knowledge, attitudes and practices of community treatment supporters administering multidrug-resistant tuberculosis injections: a cross-sectional study in rural Eswatini 

Dear Dr. Peresu:

I'm pleased to inform you that your manuscript has been deemed suitable for publication in PLOS ONE. Congratulations! Your manuscript is now with our production department. 

Kind regards, 

on behalf of

Dr. Bassey E. Ebenso 

Academic Editor

PLOS ONE